Subject Area:
immunology

Keywords:
VHH, cancer vaccines, cytokines, cancer immunology, neoantigens, alpaca nanobodies

Author for correspondence:
Stephanie K. Dougan
e-mail: stephanie_dougan@dfci.harvard.edu

†These authors contributed equally to this study.

‡Deceased 18 April 2018.

# Neoleukin-2 enhances anti-tumour immunity downstream of peptide vaccination targeted by an anti-MHC class II VHH

Stephanie J. Crowley[1,†], Patrick T. Bruck[1,†], Md Aladdin Bhuiyan[1,2], Amelia Mitchell-Gears[1,3], Michael J. Walsh[4], Kevin Zhangxu[1], Lestat R. Ali[1], Hee-Jin Jeong[1,5], Jessica R. Ingram[1,‡], David M. Knipe[4], Hidde L. Ploegh[6], Michael Dougan[2] and Stephanie K. Dougan[1,7]

[1]Department of Cancer Immunology and Virology, Dana-Farber Cancer Institute, Boston, MA, USA
[2]Department of Medicine, Division of Gastroenterology, Massachusetts General Hospital and Harvard Medical School, Boston, MA, USA
[3]University of Leeds, Leeds, West Yorkshire, UK
[4]Program in Virology and Department of Microbiology, Harvard Medical School, Boston, MA, USA
[5]Department of Biological and Chemical Engineering, Hongik University, Mapo-gu, Seoul, Korea
[6]Program in Cellular and Molecular Medicine, Boston Children's Hospital, Boston, MA, USA
[7]Department of Immunology, Harvard Medical School, Boston, MA, USA

DMK, 0000-0003-1554-6236; SKD, 0000-0002-2263-363X

Cancer-specific mutations can lead to peptides of unique sequence presented on MHC class I to CD8 T cells. These neoantigens can be potent tumour-rejection antigens, appear to be the driving force behind responsiveness to anti-CTLA-4 and anti-PD1/L1-based therapies and have been used to develop personalized vaccines. The platform for delivering neoantigen-based vaccines has varied, and further optimization of both platform and adjuvant will be necessary to achieve scalable vaccine products that are therapeutically effective at a reasonable cost. Here, we developed a platform for testing potential CD8 T cell tumour vaccine candidates. We used a high-affinity alpaca-derived VHH against MHC class II to deliver peptides to professional antigen-presenting cells. We show *in vitro* and *in vivo* that peptides derived from the model antigen ovalbumin are better able to activate naive ovalbumin-specific CD8 T cells when conjugated to an MHC class II-specific VHH when compared with an irrelevant control VHH. We then used the VHH-peptide platform to evaluate a panel of candidate neoantigens *in vivo* in a mouse model of pancreatic cancer. None of the candidate neoantigens tested led to protection from tumour challenge; however, we were able to show vaccine-induced CD8 T cell responses to a melanoma self-antigen that was augmented by combination therapy with the synthetic cytokine mimetic Neo2/15.

## 1. Introduction

CD8 T cells can recognize tumours via cancer antigens presented on MHC class I. These cancer antigens come in several categories, including developmental or tissue-restricted antigens, self-antigens with altered post-translational modifications and viral antigens in the case of viral-associated cancers [1]. In addition, mutations acquired during the process of oncogenesis can lead to altered peptide sequences presented on MHC class I and class II. These so-called neoantigens can be potent tumour-rejection antigens and appear to be the driving force behind responsiveness to anti-CTLA-4 and anti-PD1/L1-based therapies [2,3]. Given the unique specificity for tumour versus healthy

tissue and the lower degree of tolerance induction, neoantigen-based vaccines are currently in clinical trials for several types of advanced malignancies [4]. The platform for delivering neoantigen-based vaccines has varied, with both RNA and synthetic long peptides being equally effective in small-scale trials [5–8]. The adjuvants used have been largely empirical and have been selected based on safety profile and availability rather than optimal stimulation of tumour-specific CD8 T cell responses. Further optimization of both platform and adjuvant will be necessary to achieve scalable vaccine products that are therapeutically effective at a reasonable cost [9].

Recently activated T cells require IL-2 signalling to sustain their growth and proliferation [10]. Recombinant IL-2 is approved as a therapy for melanoma but is limited by severe systemic toxicity and the preferential induction of proliferation of Foxp3+ regulatory T cells (Tregs) that express the high-affinity IL2Rα chain CD25 [11]. High-dose IL-2 is required to overcome the sink effect of Treg-expressed CD25; however, these doses induce severe systemic side effects requiring hospitalization [12]. Structural variants of IL-2 that selectively bind the IL2Rβγ complex without binding to IL2Rα have been developed, although like naturally occurring cytokines, these variants suffer from limited thermal stability [13–16]. A synthetically designed protein called Neoleukin-2 (Neo2/15) was reported to exclusively bind IL2Rβγ without binding to IL2Rα [17]. This engineered cytokine displayed improved thermal stability and was able to induce T-cell proliferation *in vitro* even after boiling the cytokine for an hour [17]. Neo2/15 augmented the therapeutic efficacy of the melanoma-specific antibody TA99 in a preclinical model and had a lower toxicity profile compared with recombinant murine IL-2 [17]. We therefore tested whether Neo2/15 could be used to augment peptide vaccine-induced CD8 T cell responses in a similar model.

Most conventional vaccine strategies elicit neutralizing antibody responses but fail to generate antigen-specific CD8 T cells. To prime naive T-cell responses, the antigen must be expressed by or targeted to a professional antigen-presenting cell (APC). Several methodologies have been used to address this challenge including injection of DNA or RNA into the skin, use of live viral vectors or loading of dendritic cells *ex vivo*. Delivery of antigen to professional APCs can also be achieved by targeting unique cell surface receptors. To this end, we reasoned that MHC class II is constitutively expressed on professional APCs and is endocytosed, leading to localization of bound cargo to the endolysosomal compartment [18,19]. Alpaca nanobody fragments (VHHs) against MHC class II have been used to target antigenic cargo for endocytic processing and presentation on MHC class II to CD4 T cells [19–21]. Although cross-presentation of these same cargos on MHC class I for activation of CD8 T cells was somewhat limited, the lower degree of CD8 T cell priming could reflect the modest affinity of the anti-MHC class II VHH7 for its target [19,22]. DC15 is a VHH specific for MHC class II that binds with fivefold higher affinity than VHH7 and can competitively inhibit VHH7 binding to its target [22]. We therefore evaluated whether conjugation of antigenic peptides to the higher-affinity anti-MHC class II VHH DC15 would be capable of eliciting CD8 T cell priming.

Here, we evaluate a novel strategy of using MHC class II expression to target CD8 T cell epitopes to professional APCs. We further evaluate the combination with Neo-2/15 and

show augmentation of CD8 T cell responses in a preclinical model of melanoma.

## 2. Results

We expressed the high-affinity anti-MHC class II VHH clone DC15 or a control VHH clone 96G3 m (VHHcont) with an LPETGG sortase recognition motif at the C-terminus [22–24]. Sortase was used to install GGG-TAMRA, and the resultant fluorescently labelled DC15 was shown to bind to MHC class II positive B cells by flow cytometry, validating proper expression and folding (figure 1a). Antigenic peptides were synthesized with N-terminal triglycine motifs ($G_3$) and linked to the VHHs using recombinant 7+ sortase (figure 1b). Peptides also contained a biotin tag for detection of properly conjugated VHHs as determined by immunoblot with streptavidin–HRP and detection of a biotin-containing protein at 15 kDa (figure 1c). The concentration of properly conjugated VHH-peptide was determined by anti-biotin ELISA (figure 1d). Anti-biotin ELISA provides a quantitative readout of properly conjugated material, and this method was used subsequently to determine the concentration of VHH-peptide conjugates.

To determine whether conjugation to DC15 enhanced presentation of antigenic peptides on MHC class I, DC15 or VHHcont were conjugated to SIINFEKL peptide, the ovalbumin epitope recognized by CD8 T cells from OT-I transgenic mice [25]. VHH-peptide conjugates or molar equivalents of free VHH admixed with SIINFEKL were pulsed onto anti-CD40-activated B-cell blasts (APCs) for 30 min. APCs were washed and cocultured with OT-I T cells. CD8 T cell activation was measured by multiple parameters including proliferation, production of IFNγ and upregulation of the activation markers CD69 and CD25 (figure 2). Importantly, the amounts of peptide used in these cocultures were below that required for the activation of OT-I T cells by surface loading onto MHC class I, as evidenced by minimal activation induced by DC15 admixed with free peptide at concentrations lower than 300 pM (figure 2, blue bars).

We next assessed whether DC15-conjugated SIINFEKL could activate naive OT-I T cells *in vivo* better than peptides conjugated to an irrelevant control VHH. To this end, we injected equimolar amounts of DC15-SIIN or VHHcont-SIIN into the left foot pad of C57BL/6 mice that had received CFSE-labelled naive OT-I T cells by adoptive transfer. Contralateral footpads were injected with PBS to provide an internal negative control for each mouse. Popliteal lymph nodes were harvested 3 days later, and proliferation indexes were calculated based on CFSE dye dilution of proliferating OT-I T cells. At both 2 and 10 ng doses of vaccine, DC15 conjugation induced superior CD8 T cell activation compared with VHHcont (figure 3a). This effect was dependent on MHC class II expression, as DC15-SIIN was less effective in mice genetically deficient in MHC class II compared with wild-type controls (figure 3b).

MHC class II is expressed by multiple cell types, including B cells, macrophages and dendritic cells. Of these, B cells are by far the most abundant cell type in lymph nodes, and several groups have proposed that B cells serve as an antigen sink, given their inability to prime naive CD8 T cells [26–28]. CD8 T cells are instead primed through interactions with dendritic cells, with

royalsocietypublishing.org/journal/rsob    Open Biol. **10**: 190235

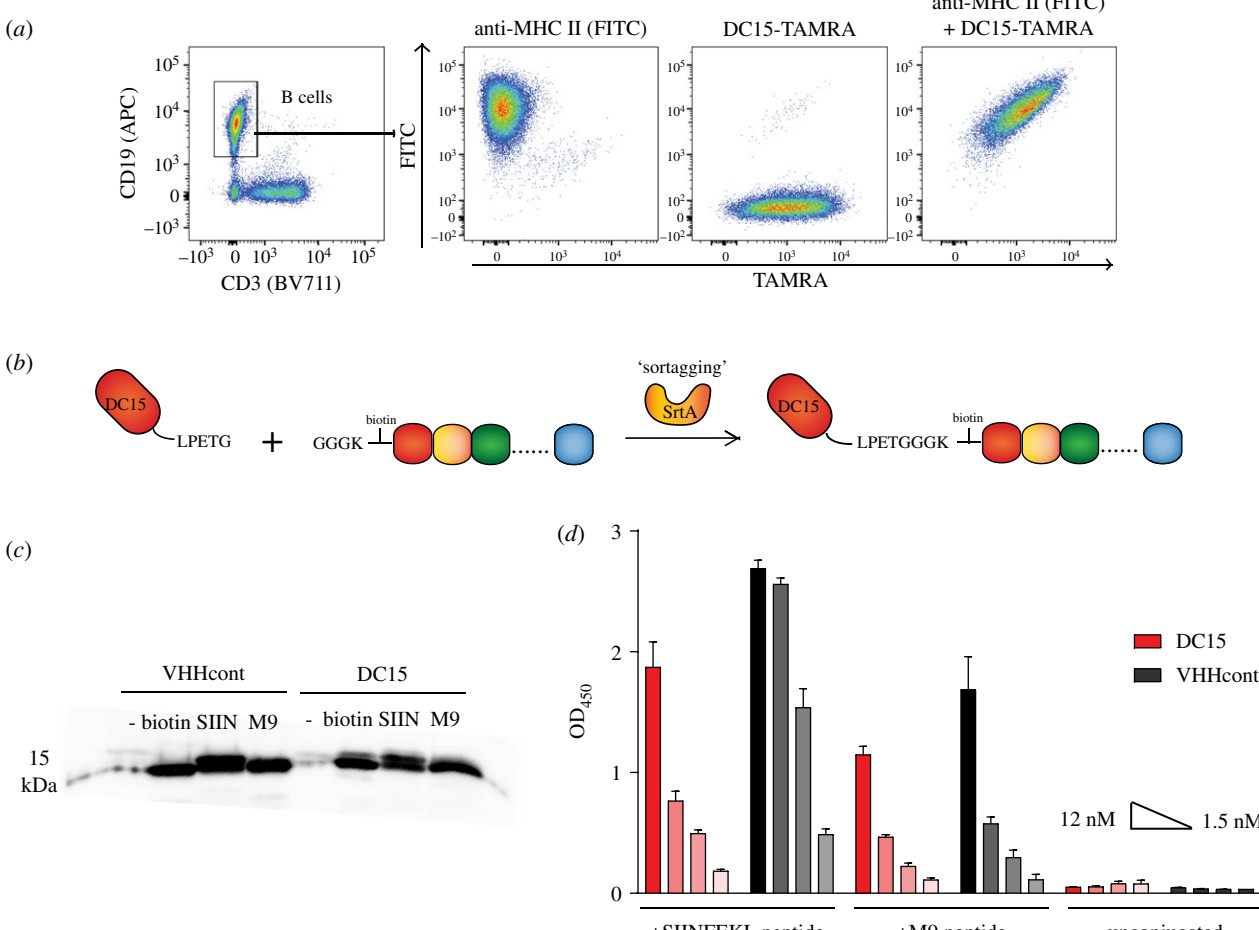

**Figure 1.** Construction of peptide vaccine conjugated to anti-MHC class II nanobody DC15. (*a*) Mouse spleen cells were stained with antibodies to CD3, CD19, MHC class II and DC15-TAMRA as indicated and analysed by flow cytometry. DC15-TAMRA and anti-MHC class II were used at equimolar ratios. (*b*) Scheme for production of antigen-loaded DC15. The MHC class II-specific VHH DC15 is expressed with a C-terminal LPETGG sortase recognition motif. Antigenic peptides are synthesized with an N-terminal triglycine motif for sortase-mediated conjugation to DC15. Multiple peptide epitopes may also be linked in tandem array. (*c*) VHHs and VHH conjugates were analysed by SDS–PAGE followed by transfer to nitrocellulose membrane and analysis with streptavidin–HRP. (*d*) Anti-biotin ELISA was performed on titrated samples of VHH and VHH conjugates as indicated. Peptides used were from ovalbumin (SIINFEKL) or the melanoma antigen TRP1 (M9).

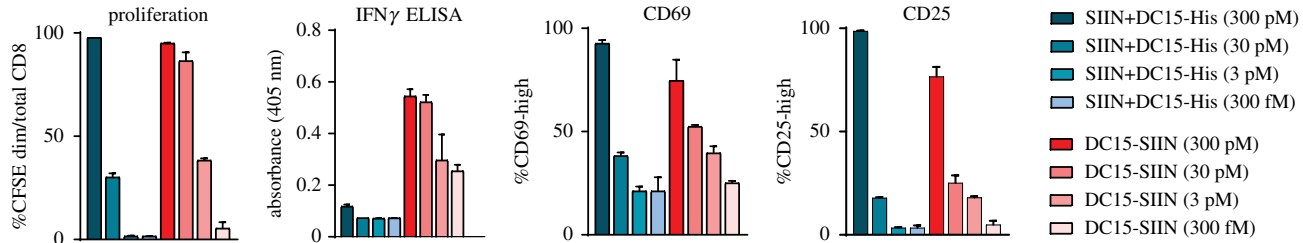

**Figure 2.** DC15 targeting increases antigen-presentation *in vitro*. B cells were isolated from mouse spleen by negative selection using CD43 magnetic beads and cultured with agonistic anti-CD40 for 3 days to induce B-cell blasts for use as APCs. APCs were pulsed for 30 min with the indicated concentrations of DC15 either admixed with (blue bars) or directly conjugated to SIINFEKL peptide (red bars). APCs were then washed to remove any unbound or non-internalized antigen and cocultured with CFSE-labelled OT-I T cells. Proliferation was measured by flow cytometry after 72 h. IFNγ was measured by ELISA of 72 h culture supernatants. CD69 and CD25 were measured by flow cytometry at 24 h from replicate cultures. Representative of three independent experiments. Error bars are s.e.m.

Batf3+ CD103+ cross-presenting cells showing superior T-cell priming compared with other DC subsets [29]. To determine whether DC15-SIIN targeting is affected by the presence of a B-cell sink, we vaccinated μMT−/− mice that lack peripheral B cells [30]. DC15-SIIN vaccination elicited similar OT-I T-cell responses in both wild-type and μMT−/− mice (figure 3*c*), suggesting that MHC class II+ dendritic cells are more likely the relevant APC in this setting. We confirmed that cross-presentation of SIINFEKL peptide on MHC class I is required, as vaccination responses failed to be elicited in β2m−/− mice lacking expression of MHC class I (figure 3*c*). Collectively, these

experiments demonstrate that targeting of antigenic peptide to MHC class II+ cells *in vivo* elicits CD8 T cell priming, likely through the conventional pathway of cross-presentation on MHC class I by specialized dendritic cells.

DC15 can be easily conjugated to a variety of peptides, and we hypothesized that this platform could be used for neoantigen vaccines in cancer. To test this, we used a pancreatic cancer cell line KPC.1 derived from a spontaneously arising tumour from a *LSL-Kras*^G12D;*p53*+/*flox,p48-cre* mouse [31]. The donor mouse was 95% C57BL/6 background and matched for MHC haplotype. However, 5% of non-C57BL/

royalsocietypublishing.org/journal/rsob    Open Biol. **10**: 190235

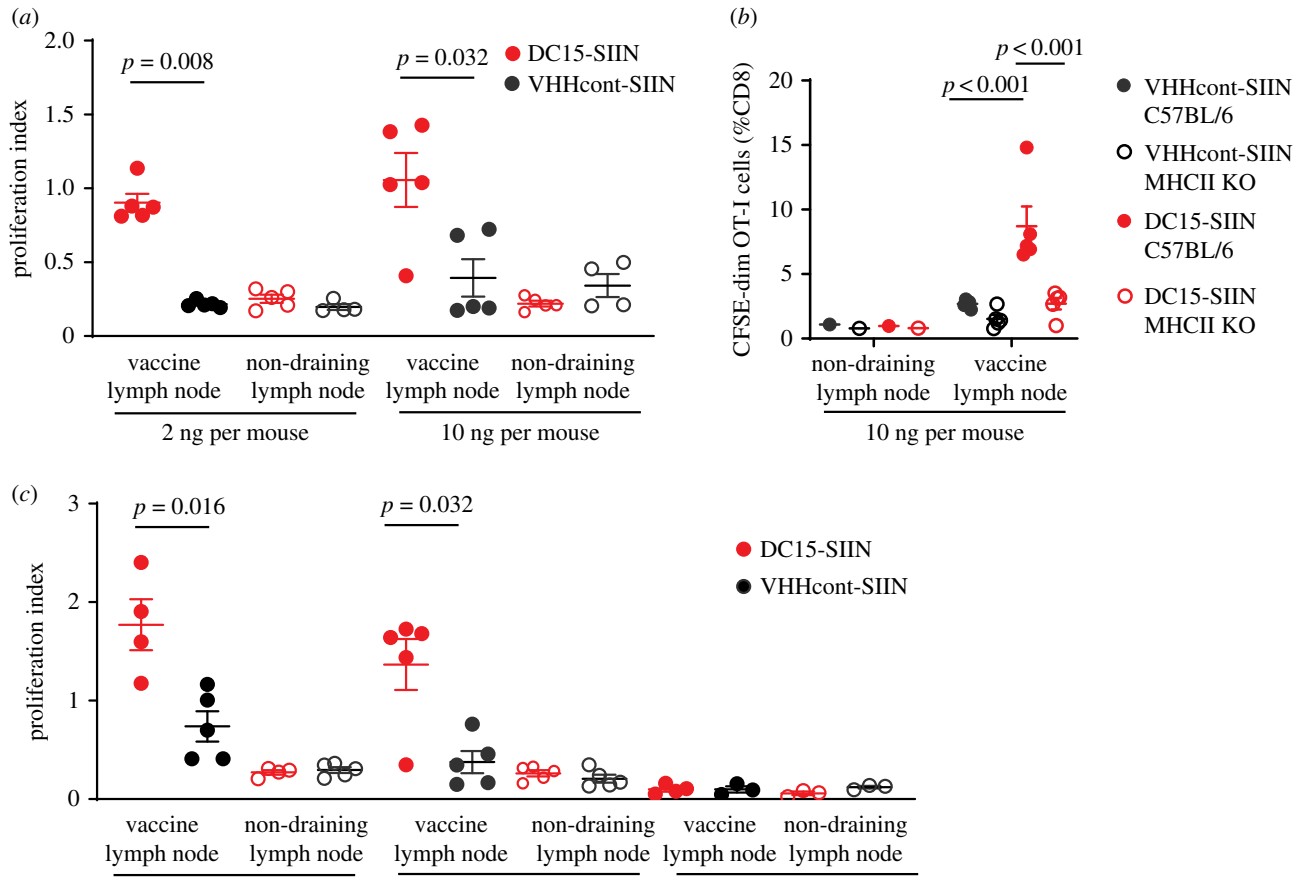

**Figure 3.** DC15 targeting increases CD8 T cell priming *in vivo* at low doses and does not require B cells. Pooled spleen and lymph node cells from OT-I mice were labelled with CFSE and transferred intravenously into C57BL/6 recipients at $10^6$ cells per mouse. (*a*) Mice were then immunized in the left footpad with DC15-SIIN or VHHcont-SIIN at 2 or 10 ng per mouse. Right footpads were injected with PBS. Popliteal lymph node cells were collected 72 h later and analysed by flow cytometry. Proliferation index was calculated as the ratio of mitotic events to progenitor cells. Representative of four independent experiments. (*b*) OT-I cells were transferred into C57BL/6 or MHC class II-deficient hosts. Host mice were immunized with 10 ng per mouse DC15-SIIN or VHHcont-SIIN. Proliferation was measured as the percentage of CFSE-dim Vα2+ Vβ5+ OT-I cells out of total CD8+. (*c*) OT-I cells were transferred into C57BL/6, μMT−/− or β2m−/− hosts. Host mice were immunized with 10 ng per mouse DC15-SIIN or VHHcont-SIIN. Proliferation indexes were calculated as in (*a*). Representative of two independent experiments.

**Table 1.** Model neoantigens from KPC.1 cells used in this study.

|  | MHC I | gene | epitope sequence | netMHC score | mRNA expression (FPKM) |
|---|---|---|---|---|---|
| Neo1 | H-2-Db | Lars | NMIEAGDAL | 1.5 | 6943 |
| Neo2 | H-2-Kb | Hjurp | VSALSSRV | 1.75 | 3608 |
| Neo3 | H-2-Db | Smcr8 | RALRKQQPI | 0.2 | 1838 |
| Neo4 | H-2-Db | Smcr8 | VSIPPQSYI | 1.3 | 1838 |
| Neo5 | H-2-Kb | Kntc1 | TGLRFHEL | 0.24 | 1712 |
| Neo6 | H-2-Kb | Cdt1 | MSYRFRQE | 0.17 | 1711 |
| Neo7 | H-2-Db | Cdt1 | GQIKTVYPM | 0.9 | 1711 |
| Neo8 | H-2-Db | Cdt1 | EMFHSMDTI | 2 | 1711 |
| Neo9 | H-2-Db | Slc9a1 | PSLLMVVAL | 1.9 | 954 |
| Neo10 | H-2-Kb | Gadd45gip1 | SGVLPASL | 1.7 | 938 |
| Neo11 | H-2-Kb | Ppp1r21r | KLRTYVTL | 1.7 | 745 |

6 contributes to approximately 1000 SNPs. We used IEDB to identify putative MHC class I binding epitopes and used these as model neoantigens. Putative model neoantigens were ranked based on the likelihood of binding to MHC class I ($K^b$ or $D^b$) and their relative expression level in cultured KPC.1 cells by RNAseq analysis (table 1). The top 11 model neoantigens were synthesized with triglycine motifs

and biotin at the N-termini, and sortase was used to conjugate them to DC15 or VHHcont (figure 4a). Two peptides (neo8 and neo9) had poor solubility and were not analysed further. The remaining VHH-peptides were pooled and used to vaccinate C57BL/6 mice at days 14 and 7 prior to inoculation with subcutaneous KPC.1 cells. Unfortunately, no differences were observed between the rate of growth or

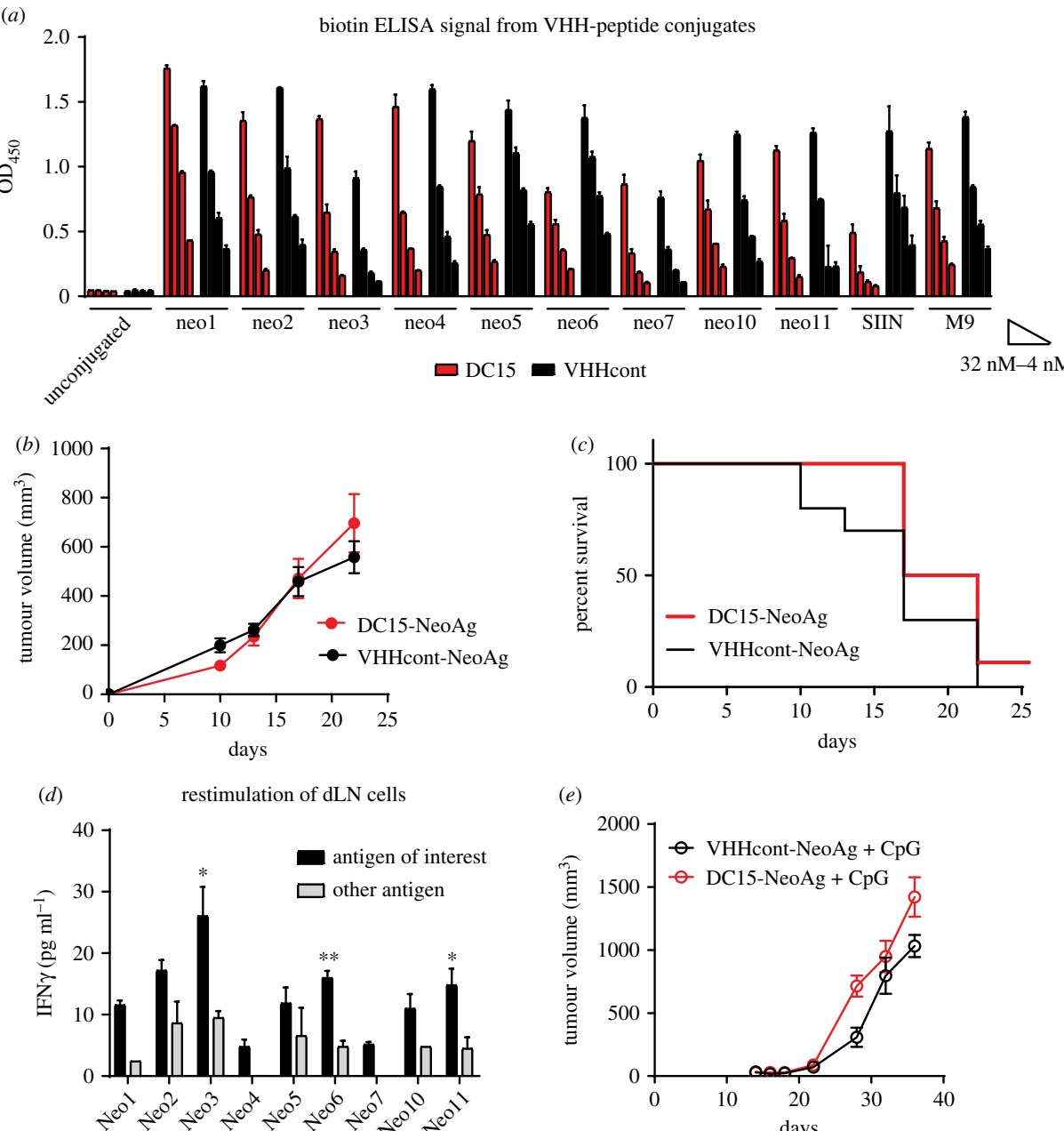

**Figure 4.** DC15-based neoantigen vaccine in pancreatic cancer is ineffective at inducing T-cell responses. (a) Neoantigens from table 1 were synthesized as bio-tinylated 18-mer peptides containing the 8- or 9-mer neoantigen with three or four flanking residues from the original gene and GGG motifs at their N-termini. Peptides were conjugated to DC15 or VHHcont using sortase. Conjugation was validated by anti-biotin ELISA. (b) All nine neoantigens were pooled at 2 ng per VHH-neo construct per mouse per immunization (18 ng total protein per injected paw). Mice were immunized in both hind footpads at days 14 and 7 prior to inoculation of 250 000 KPC.1 cells subcutaneously. Tumour growth was monitored over time. $n = 10$ mice per group. (c) Survival of mice shown in (b). (d) C57BL/6 mice were vaccinated as in (b). Draining popliteal LNs were harvested 7 days after the second immunization, and dissociated cells were cultured with 1 ng ml$^{-1}$ of the indicated neoantigen peptides or irrelevant SIINFEKL control. IFN$\gamma$ was measured by ELISA of 48 h culture supernatants. $*p < 0.02$, $**p < 0.001$. Error bars are s.e.m. of biological duplicates. (e) Mice were vaccinated as in (b) except that 15 ng CpG was admixed with the VHH conjugates prior to inoculation. Mice were challenged with 250 000 KPC.1 cells subcutaneously. Tumour growth was monitored over time. $n = 5$ mice per group.

overall survival of tumour-bearing mice (figure 4b,c). To determine whether DC15-NeoAg pool induced antigen-specific T cells, a cohort of non-tumour-bearing C57BL/6 mice were vaccinated at days 14 and 7 prior to harvest of vaccine draining LN cells and restimulation *ex vivo* with each model neoantigen (figure 4d). Although most of the peptides induced greater IFN$\gamma$ production than their irrelevant controls, the overall levels of IFN$\gamma$ production were low, consistent with lack of a vigorous neoantigen-specific T-cell response. We hypothesized that the addition of the TLR9 ligand CpG would adjuvant the DC15-NeoAg response; however, even with the addition of CpG, tumours grew

progressively in both VHHcont-NeoAg and DC15-NeoAg vaccinated mice (figure 4e).

Pancreatic cancer is notoriously refractory to CD8 T cell-based therapies [32]. To test the effects of DC15-peptide vaccination in a more amenable setting, we used the B16 melanoma model. Tyrosinase-related protein 1 (TRP1) is a CD8 T cell antigen in both mice and humans, and TRP1-specific CD8 T cells can be tracked using specific tetramers [33]. We conjugated DC15 or VHHcont to the TRP1 peptide M9 (TAPDNLGYM), which has an alanine to methionine substitution in the ninth position anchor residue to enhance the stability of the H-2D$^b$ peptide complex [33,34]. C57BL/6

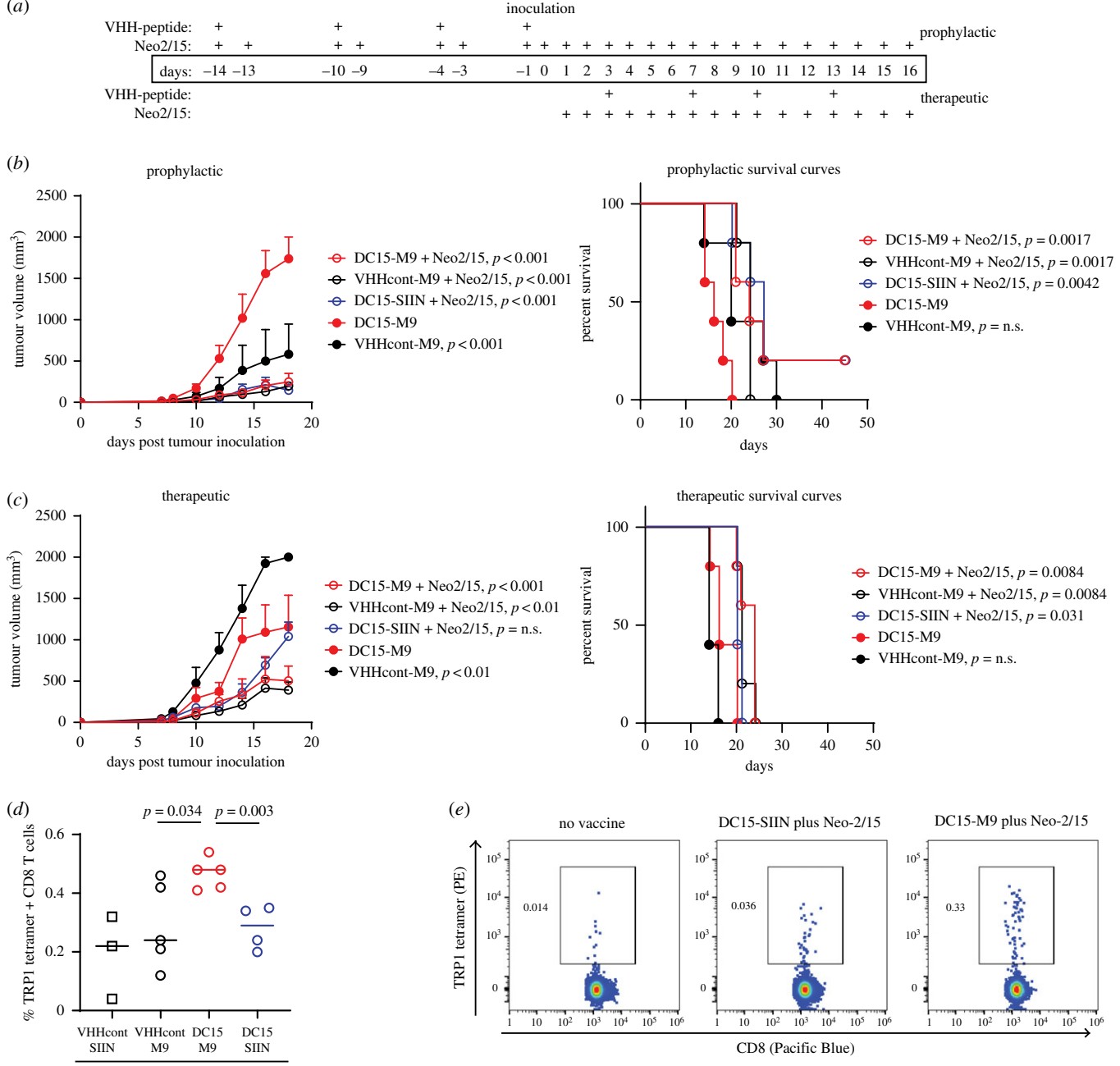

**Figure 5.** Combination of DC15-based vaccination and Neo2/15 leads to anti-tumour immunity in the B16 melanoma model. (*a*) Schedule of treatments for prophylactic and therapeutic experimental cohorts. Neo2/15 was used at 10 μg per dose; VHH-peptides were used at 10 ng per dose. M9 peptide was used as the relevant antigenic epitope; SIINFEKL was included as a negative control. $n = 5$ mice per group. Experiments were conducted in parallel such that all mice were inoculated subcutaneously with 250 000 B16 cells on the same day. (*b*) Tumour growth and survival of mice in the prophylactic cohorts. *p*-values shown are comparisons with the DC15-M9 vaccinated group. (*c*) Tumour growth and survival of mice in the therapeutic cohorts. *p*-values shown are comparisons with the DC15-M9 vaccinated group. (*d*) C57BL/6 mice were vaccinated with 10 ng per dose of VHH-peptide on days 0 and 4. Mice received 10 μg per mouse Neo2/15 daily starting on day 0. Draining lymph nodes were harvested on day 9, stained and analysed by flow cytometry. (*e*) C57BL/6 mice were inoculated with B16 tumours and left untreated. When tumours reached greater than 1 cm³, draining lymph node cells were harvested from mice with progressively growing tumours as well as from the single surviving mice in the DC15-SIIN/Neo2/15 and DC15-M9/Neo2/5 prophylactic cohorts shown in (*b*). Cells were stained with anti-CD8 and TRP1 tetramers. Flow cytometry plots are gated on total CD8+ cells.

mice were vaccinated with 10 ng per dose of DC15-M9, VHHcont-M9 or DC15-SIIN (used here as an irrelevant control peptide) according to the schedule shown in figure 5*a*. Two experiments were performed simultaneously, one in which mice were vaccinated prior to tumour challenge (prophylactic) and one in which mice were vaccinated starting 3 days after tumour challenge (therapeutic). Given the poor adjuvant effects observed with CpG (figure 4*e*), we decided to use the synthetic IL-2 mimetic Neo2/15 instead. Both IL-2 and Neo2/15 were previously shown to have modest single-agent activity and to augment responses to TA99, a TRP1-specific antibody, in the B16 model [17,35].

We observed a significant decrease in tumour growth in all cohorts that received Neo2/15, consistent with the known single-agent activity of this compound (figure 5*b*,*c*). All mice were inoculated with B16 tumours on the same

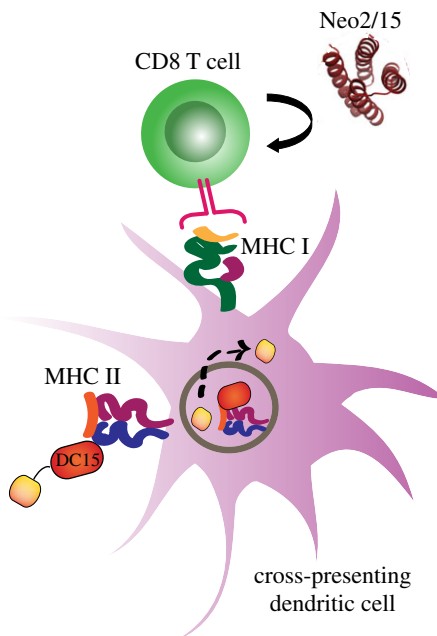

**Figure 6.** Diagram of tumour vaccine platform. DC15 binds to MHC class II on the surface of dendritic cells, which is internalized into endolysosomes, allowing for the proteolytic release of the antigenic peptide cargo (yellow). The antigenic peptide is then cross-presented on MHC class I to CD8 T cells. Neo2/15 supports the proliferation and survival of newly primed CD8 T cells.

day, allowing for comparison across all cohorts. Intriguingly, the tumour growth delay induced by Neo2/15 was more pronounced in the prophylactic setting, suggesting that tonic low-level T-cell activation at the time of tumour implantation may provide some protective benefit. We did not observe a significant difference in tumour control or survival between mice that received DC15-M9 versus VHHcont-M9, and indeed, the DC15-M9 prophylactic cohort showed increased tumour growth compared with VHHcont-M9. We speculate that this could be due to DC15 binding to B cells expressing MHC class II. However, in the therapeutic cohort, DC15-M9 vaccination performed better than VHHcont-M9. Inoculation of the tumour itself acts as a complex mixture of vaccine antigens and it could be that priming with a single peptide followed by boosting with a complex mixture is less effective than priming with a complex mixture and boosting with a single peptide [36].

To ascertain whether DC15-M9 vaccination induced any TRP1-specific CD8 T cells, we first evaluated mice treated according to the prophylactic schedule shown in figure 5*a*, but harvested at day 5. These mice received two doses of vaccine and Neo2/15 but were not challenged with the tumour. At this early time point, we observed that DC15-M9 vaccinated mice had TRP1 tetramer+ CD8 T cells at a frequency above the baseline level seen in mice vaccinated with SIINFEKL irrelevant peptide (figure 5*d*). To determine whether vaccine-elicited CD8 T cells were observed in mice that had survived tumour challenge, we harvested draining lymph nodes from the two surviving tumour-free mice, one that had received DC15-SIIN/Neo2/15 and one that had received DC15-M9/Neo2/15. TRP1 tetramer staining showed a 10-fold increase in TRP1-specific CD8 T cells in the mouse that had received DC15-M9 vaccination compared with the mouse that had received DC15-SIIN (figure 5*e*). These results suggested that, while Neo2/15 is capable of augmenting a

polyclonal endogenous response to B16 tumours, in DC15-M9 vaccinated mice the vaccine-elicited CD8 T cells may also contribute to tumour control (figure 6).

## 3. Discussion

Cancer vaccines have been fraught with challenges, and the field is only now beginning to see successful phase I trials after nearly three decades of effort [37,38]. A successful vaccine must incorporate three parts: high-quality antigens, potent adjuvants and a robust delivery platform. Here, we present a novel approach, namely, using a high-affinity MHC class II-specific VHH to target antigenic peptides to antigen-presenting cells. This strategy was effective using as little as 2 ng of protein conjugate to elicit CD8 T cell responses *in vivo* to the model antigen SIINFEKL. Other antigens could be easily conjugated through the sortase handle installed on the VHH, enabling us to rapidly evaluate a panel of neoantigen candidates in a pancreatic cancer model. Although none of these candidates afforded tumour protection, the ease of the delivery platform allowed us to test and eliminate non-productive candidates quickly. Tandem addition of peptide epitopes is also trivial, either through direct genetic linkage or through repeat sortagging of individual subunits. Although DC15 is specific for mouse MHC class II, a human-specific version that recognizes multiple HLA-DR haplotypes has been reported [39].

Alpaca-derived VHHs have shown utility in preclinical models of cancer diagnostics and therapy [21]. In addition to MHC class II, other endocytosed receptors have been targeted, and CD11b-specific VHHs are capable of generating CD8 T cell responses [40]. VHHs have also been used to deliver immunomodulatory cytokines to the tumour microenvironment [41], and as immune-positron emission imaging reagents for cancer and targets of cancer therapies [40,42–45]. Due to their unusually stable folding, single-domain VHHs generate stable CAR-T constructs and can be used to target solid tumours in mice [46]. Delivery of increasingly complex vaccine cargo is a reasonable next step, perhaps aided by the selection of VHHs specific for targets more exclusively restricted to cross-presenting dendritic cells. MHC class II targeted VHHs have been used as vaccines for both infectious disease and cancer in settings where CD4T cell responses and antibody responses conferred protective immunity [19,20]. They have thus far been less good at eliciting CD8 T cell responses, although we have now demonstrated proof-of-principle efficacy using the high-affinity anti-MHC class II targeted VHH clone DC15.

We chose to support newly primed CD8 T cell responses using the synthetic IL-2 mimetic Neo2/15. This highly stable protein is incapable of binding to IL2Ra (CD25), giving it a favourable safety profile [17]. Importantly, Treg induction is less pronounced than with regular IL-2, and a direct comparison of Neo2/15 with equimolar recombinant IL-2 showed greater anti-tumour activity of the synthetic cytokine [31]. Here, we also observed significant activity of Neo2/15 even when combined with DC15 conjugated to an irrelevant peptide. These results highlight the importance of a polyclonal response to achieving successful tumour control [47]. Neo2/15 was also able to support vaccine-elicited CD8 T cells, as evidenced by increased TRP1 tetramer+ cells early in the vaccination protocol and in the surviving mouse that

had received DC15-M9 vaccination. We are moderately encouraged that the DC15 platform can be used to prime CD8 T cell responses in the setting of cancer.

# 4. Material and methods

## 4.1. Mice

All animal protocols were approved by the Dana-Farber Cancer Institute Committee on Animal Care (protocol nos 14-019 and 14-037) and are in compliance with the NIH/NCI ethical guidelines for tumour-bearing animals. The following mouse strains were purchased from Jackson Labs: C57BL/6 (000664), μMT$^{-/-}$ (002249), β2m$^{-/-}$ (002087), OT-I (003831) and I-Ab$^{-/-}$ (005589).

## 4.2. Subcutaneous tumour inoculations

The B16F10 cell line was purchased from ATCC and used within 1 year of receipt. The KPC.1 cells were a gift from A. Maitra (MD Anderson Cancer Center). Cells were cultured in complete RPMI media and were verified by Charles River Laboratories to be mouse pathogen free and mycoplasma free less than six months prior to use. Cells were grown to 80% confluency, dissociated using 0.25% trypsin, washed twice in PBS and suspended in fresh sterile PBS for inoculation into mice. Mice were inoculated with 250 000 tumour cells in 150 μl total volume. Tumour size was monitored by precision calipers every 2–3 days. Mice were euthanized when tumour size reached 2 cm$^3$, ulcerated or showed signs of morbidity consistent with the NIH/NCI ethical guidelines for tumour-bearing animals.

## 4.3. OT-I cell transfer and footpad vaccinations

Spleen and lymph nodes cells were isolated from 2–3 donor OT-I mice, subjected to hypotonic lysis to remove erythrocytes, and labelled with CFSE (Invitrogen, C34554) according to the manufacturer's protocol. CFSE-labelled cells were washed twice in PBS, counted and suspended at 1 million cells per 150 μl sterile PBS. Cells were then transferred by tail vein injection into host mice (1 million cells per mouse). Within 24 h, host mice were vaccinated by intradermal injection of 30 μl sterile PBS or VHH conjugates with or without CpG (Invivogen, tlrl-1826, 20 μg mouse$^{-1}$) into the hind footpads. Three days later, mice were euthanized by CO$_2$ inhalation and popliteal lymph nodes were harvested. Lymph node cells were stained with anti-CD8 and TRP1 tetramer (NIH tetramer core facility) and analysed by flow cytometry using a Sony SP6800 spectral flow cytometer. CFSE-dim cells were gated by the number of cell divisions and the proliferation indexes were calculated.

## 4.4. Cell culturing

Primary cells and cell lines were cultured in RPMI 1640 medium (Gibco, 11875119) supplemented with 10% heat-inactivated FBS (Omega Scientific catalogue no. FB-11), 2 mM L-glutamine (Gibco), 100 U ml$^{-1}$ penicillin-streptomycin (Gibco), 1 mM sodium pyruvate (Gibco), 0.1 mM nonessential amino acids (Gibco) and 0.1 mM β-mercaptoethanol (Sigma). B cells were isolated from C57BL/6 mouse spleen and lymph nodes using magnetic bead enrichment (Thermo Fisher Dynabeads Mouse CD43, 11422D) and stimulated with agonistic anti-CD40 (clone

HM40-3, 2 μg ml$^{-1}$ BD Cell Analysis, 553721). For cytokine analysis, CD40-activated B cells and CD8$^+$ T cells were cocultured at a 1 : 1 ratio, and supernatants were harvested after 72 h. IFNγ was quantified by ELISA (Biolegend, 430806). Supernatants were diluted 1 : 4 prior to analysis and used at 100 μL volume per well of a 96-well plate. For culturing of lymph node cells from vaccinated mice in figure 4, culture supernatants were not diluted prior to ELISA.

## 4.5. Flow cytometry

Cells were incubated with antibody staining mix including 2% fetal calf serum for 30 min at 4°C, washed once in PBS and resuspended in 1% formalin in PBS. The analysis was performed on a Sony SP6800 spectral flow cytometer. Data were analysed using FloJo software. Cells were first gated on CD45$^+$ cells using SSClow as a proxy for viability. Flow cytometry antibodies used in this study were purchased from Biolegend (CD8 (clone 53-6.7), CD25 (clone 3C7) and CD69 (clone H1.2F3)). TRP1 H-2D$^b$ tetramer was provided by the NIH Tetramer Core Facility. Tetramer staining was performed at room temperature for 30 min.

## 4.6. Expression and purification of sortase A

BL21 (DE3) cells were transformed with pET30b+ containing 7+ SrtA construct [23] and cultured at 37°C overnight in 5 ml of Luria Broth media supplemented 34 μg ml$^{-1}$ kanamycin. This was then was used to inoculate 200 ml of terrific broth (TB) media supplemented with 34 μg ml$^{-1}$ kanamycin (Sigma, K4000) and cultured at 37°C until and OD600 approximately 0.6, at which point 1 mM isopropylthio-β-galactopyranoside (IPTG, Teknova T0918) was added and cultures induced overnight at 30°C. Cells were harvested by centrifugation (6000 rpm, 30 min, 4°C) and the resulting pellet was resuspended in 50 ml of wash buffer (50 mM Tris, 150 mM NaCl, 10 mM imidazole, pH 7.6) and lysed by sonication. To harvest the soluble fraction, the lysate was again centrifuged (6000 rpm, 30 min, 4°C) and the resulting supernatant was incubated with 2 ml of Ni-NTA agarose resin (Qiagen, 30230) on a rotating wheel at 4°C overnight. The resin was washed three times with 10 ml of wash buffer in a disposable gravity column. After the addition of 5 ml of elution buffer (50 mM Tris, 150 mM NaCl, 500 mM imidazole, pH 7.6), the eluent was buffer exchanged in 3 kDa MWCO ultrafiltration device (Millipore) and into 50 mM Tris (pH 7.5) and concentrated to 200 μl. Expression and purification of the protein were confirmed by SDS–PAGE analysis using 4–20% polyacrylamide gel (Bio-Rad). The concentration of the protein was calculated using A280 absorbance on a NanoDrop (Thermo).

## 4.7. Expression and purification of VHHs

WK6 cells were transformed with pHEN6-DC15 or VHHcont and cultured at 37°C overnight in 50 ml of TB (Sigma, T0918) supplemented with 100 μg ml$^{-1}$ ampicillin (Sigma, A0166) at 225 rpm. This was then used to inoculate 1 l of TB supplemented with 100 μg ml$^{-1}$ ampicillin and cultured at 37°C until OD600 approximately 0.6, at which point 1 mM IPTG (Teknova, I3325) was added and cultures induced overnight at 30°C. Cells were harvested by centrifugation (1370$g$, 15 min, 4°C). The periplasmic fraction was then released via osmotic shock by incubating the pellet in 30 ml of 1× TES

royalsocietypublishing.org/journal/rsob    Open Biol. 10: 190235

royalsocietypublishing.org/journal/rsob Open Biol. 10: 190235

buffer (0.2 M Tris, 0.65 mM EDTA, 0.5 M sucrose) on a rotating wheel at 4°C for 1 h, and then diluted with an additional 30 ml of 0.25× TES buffer, and rotated overnight at 4°C. After centrifugation (9700$g$, 15 min, 4°C), the supernatant was incubated with 2 ml of Ni$^{2+}$ NTA agarose resin (Qiagen, 30230) on a rotating wheel at 4°C for 1 h. The resins were pelleted at 325$g$ and the supernatant was collected as 'flow-through'. The resins were washed two times with 50 ml of wash buffer (50 mM Tris, pH 8, 150 mM NaCl, 10 mM imidazole) via centrifugation at 325$g$. The resins were transferred to disposable gravity-flow columns (Life Technologies, 29924) and eluted 2×with 4 ml elution buffer (50 mM Tris, 150 mM NaCl, 500 mM imidazole, pH 7.6). The eluent was buffer exchanged using 30 k MWCO ultrafiltration devices (Millipore) and into LPS-free PBS and concentrated via 10 k MWCO ultrafiltration devices (Millipore). Proteins were tested for endotoxin using a Pierce LAL chromogenic endotoxin quantitation kit (Thermo, 88282), and confirmed to be under 0.2 EU mg ml$^{-1}$. Expression and purification of the protein were confirmed by SDS–PAGE analysis using 12% polyacrylamide gel (Bio-Rad), and the concentration of the protein was measured by BCA assay (Thermo Scientific, 23235).

## 4.8. VHH conjugation using SrtA

Biotinylated peptides were synthesized by the MIT Koch Institute Biopolymers Facility. 2.5 µM SrtA and 1 mg VHH-LPETGG were added to a 5× molar excess GGG-peptide or GGG-TAMRA in 50 mM Tris and 150 mM NaCl with 10 mM CaCl$_2$. The total reaction volume was 1.7 ml. The resulting mixture was incubated at 4°C for 120 min. Unconjugated VHH was removed via Ni NTA$^{2+}$ agarose (Qiagen) incubation at 4°C for 10 min. The supernatant was collected as conjugated VHH-peptide. Conjugation efficiency was determined either by SDS–PAGE analysis and immunoblotting with streptavidin–HRP or by anti-biotin ELISA.

## 4.9. SDS–PAGE analysis and streptavidin blotting

A 15% SDS–PAGE gel was cast using National Diagnostics ProtoGel reagents (ProtoGel 30% EC-890, 4X Resolving buffer EC-892, Stacking buffer EC-893, Running buffer EC-870). 0.5 µg VHH-peptide was dyed (3X Laemmli loading dye, 1 M Tris pH 6.8, 20% SDS, glycerol, β-mercaptoethanol, bromophenol blue) and loaded on the gel and then run at 150 V until the dye front reached the bottom of the gel. The Bio-Rad Trans-Blot Turbo system (17001917) was used for the transfer of the gel to a PVDF membrane. Streptavidin–HRP (Biolegend, 405210) diluted 1 : 10 000 in 3% BSA in TBST was used for detection of biotin signal. Chemiluminescence reagent (Perkin-Elmer, NEL103E001EA) was added prior to imaging on a Bio-Rad Chem-iDoc imaging system.

## 4.10. Anti-biotin ELISA

96-well high-affinity plates (Corning, 9018) were coated with VHH-peptide in coating buffer ((3.03 g Na$_2$CO$_3$, 6 g NaHCO$_3$) l$^{-1}$, pH 9.6). Plates were left at 4°C overnight. The next day, plates were washed 3× PBST, then blocked 1 h at room temperature with PBS + 10% FBS (blocking buffer). Plates were washed 3× again with PBST, then 1 : 1000 Avidin–HRP (Biolegend, 405103) in blocking buffer was added for 1 h. Plates were washed 5× with TBST, then

developed with 100 µl TMB (Sigma, T8665). The reaction was stopped using 1 M HCl (Sigma, 258148), then read on a plate reader at absorbance 540 nm.

## 4.11. Neoantigen prediction

Neoantigen candidates were obtained by analysing SNPs found in the KPC.1 cell line via RNA sequencing. Briefly, RNA from three samples of the KPC cell line was sequenced and aligned to the mm10 genome using STAR. SNPs were called using the Broad Institute pipeline 'RNAseq short variant per-sample calling'. Only SNPs that were common to all three samples were considered. The effect of each SNP on the protein product was predicted using the tool SnpEff [48], and the resulting mutated peptide sequence was identified by overlaying on the UP000000589 mouse proteome. Candidate binding affinity to mouse MHCI was predicted using the Immune Epitope Database (IEDB) recommended 2.22 prediction method, which uses the Consensus analysis tool to combine predictions from ANN, SMM and Comblib. Neoantigen candidates were analysed as 8-mers for H-2 K$^b$ and 9-mers for H-2D$^b$ with 7 and 8 amino acids flanking each side of the SNP, respectively. The top 2% of predicted binders were selected, producing a list of 48 potential neoantigens. The binding affinity of these candidates was then compared with the binding affinity of their corresponding wild-type sequence. Neoantigen candidates were then sorted in order from the largest mean expression to the smallest. The top 11 neoantigens with the largest mean expression values which also bound better to MHCI than their wild-type peptide sequence were selected for conjugation to DC15.

## 4.12. Statistical analysis

All tumour weight and tumour infiltrates data are presented as mean with s.e.m. error bars unless otherwise noted. Significance was determined using a two-sided Mann–Whitney test to compare ranks, without assuming Gaussian distribution. For tumour growth and survival curves, significance was determined using two-way ANOVA and a log rank Mantel–Cox test, respectively. Graphpad Prism software was used to analyse data.

Ethics. All animal protocols were approved by the Dana-Farber Cancer Institute Committee on Animal Care (protocol nos14-019 and 14-037) and are in compliance with the NIH/NCI ethical guidelines for tumour-bearing animals.

Data accessibility. All data are provided in the figures and tables.

Authors' contributions. S.J.C., P.T.B., M.A.B., A.M.-G., M.J.W., K.Z. and H.-J.J. performed experiments and analysed data. P.T.B., K.Z. and L.R.A. performed computational analysis of neoantigens. J.R.I. designed the VHH-peptide conjugation strategy and provided key insights. D.M.K. and H.L.P. supervised some experiments. M.D. and S.K.D. analysed data and supervised the study. S.K.D. wrote the manuscript with input from all of the authors.

Competing interests. We declare we have no competing interests.

Funding. This work was funded by a DOD Career Development award (no. CA150378) to S.K.D., a Claudia Adams Barr Award to J.R.I. and NIH (grant no. P01 AI098681) to D.M.K.

Acknowledgements. We thank D. A. Silva and D. Baker for providing the Neo2/15. We thank the NIH Tetramer Core Facility for provision of TRP1 tetramers.

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
