## [Reviewer comments · Open Biology]

Review History

RSOB-19-0235.R0 (Original submission)

Review form: Reviewer 1

Recommendation

Accept with minor revision (please list in comments)

Do you have any ethical concerns with this paper?

No

Comments to the Author

The authors link neoantigens to an anti-MHC class II nanobody, DC15, by sortagging, and test whether these novel DC15-neoantigens can boost CD8 T cell activities against pancreatic cancer and melanoma in mice. Although anti-tumor activities of these reagents are only moderately encouraging, the idea is still worth documenting. I support the publication of this paper, but have a few comments that may help the authors improve the data quality in the paper.

1. DC15-neoantigens are initially delivered to the class II MHC compartments and cross-presented to MHC class I molecules to activate CD8 T cells. This is supported by data shown in

Figure 3B. I know $p=0.011$ is considered statistically significant; however, there are only three data points from MHCII KO mice immunized with DC15-SIIN, and one of the data point falls above the average of the five data points from similarly immunized C57BL/6 mice. The authors may consider to add a few more MHCII KO mice in this study. In fact, the authors should probably state how sample sizes in various mouse studies were determined.

2. The authors should provide statistical analyses and p values for all relevant panels in Figures 4 and 5. In Figure 5B, B16 melanoma cells grow faster in mice receiving DC15 M9 than in those receiving VHHcont M9. Is this statistically significant? If so, can the authors speculate what may have happened?

3. One single mouse was analyzed to obtain and present data shown in each flow panel in Figure 5D. The authors should repeat this experiment using appropriate mouse numbers and include statistical data to warrant data reproducibility.

Review form: Reviewer 2

Recommendation

Major revision is needed (please make suggestions in comments)

Do you have any ethical concerns with this paper?

No

Comments to the Author

Overall, the authors describe a tumor vaccination strategy that is innovative and novel. Their approach uses an MHC class II-targeting nanobody that is conjugated to a tumor peptide. Their system uses sortase to conjugate tumor peptides, making it a modular system that can be rapidly adapted for any tumor antigen or immunogenic peptide. They propose that their nanobody (DC15) binds to MHC class II on APCs and gets internalized, subsequently resulting in cross-presentation of the tumor antigen. The authors demonstrate the fundamental aspects of this strategy are valid in vitro. Unfortunately, the authors tested their vaccination strategy in two independent in vivo models and did not observe robust anti-tumor effects. They do, however, observe evidence of immune activation in vivo as a consequence of DC15 conjugated to a peptide. This is an important finding and I believe the manuscript can become outstanding with revision as described below.

Major

The manuscript would benefit from an experimental system showing definitive evidence of therapeutic efficacy in vivo. The authors show T cell stimulation in vivo using the OT-1 model. Therefore, one approach could be to evaluate anti-tumor responses with DC15-SIIN against an OVA+ tumor. Alternatively, perhaps a more robust anti-tumor response could be observed when combining a DC15-peptide conjugate with an immune checkpoint inhibitor. Along these lines, Moynihan et al 2016 Nature Med (PMID 27775706) found that peptide vaccines were most effective when combined with IL-2 analogues and immune checkpoint blockade. At the very least, the experiment reported in Figure 5D should be repeated with a larger number of animals to draw statistical conclusions. If the authors approach is not effective in models of cancer, might it alternatively work as a strategy to immunize against intracellular pathogens?

Minor

1. Has the DC15 agent been described in the literature previously? If not, the authors should present its biophysical characterization with respect to target binding and affinity. Does the protein cross-react with human MHC class II?

2. Does DC15 also stimulate CD4 T cells via direct MHC class II loading?
3. The text could benefit from a diagram that demonstrates the proposed mechanisms of action of DC15-peptide conjugates.

Decision letter (RSOB-19-0235.R0)

11-Nov-2019

Dear Dr Dougan,

We are writing to inform you that the Editor has reached a decision on your manuscript RSOB-19-0235 entitled "Neoleukin-2 enhances anti-tumor immunity downstream of peptide vaccination targeted by an anti-MHC class II VHH", submitted to Open Biology.

As you will see from the reviewers' comments below, there are a number of comments that prevent us from accepting your manuscript at this stage. The reviewers suggest, however, that a revised version could be acceptable, if you are able to address their concerns. If you think that you can deal satisfactorily with the reviewer's suggestions, we would be pleased to consider a revised manuscript.

The revision will be re-reviewed, where possible, by the original referees. As such, please submit the revised version of your manuscript within four weeks. If you do not think you will be able to meet this date please let us know.

When submitting your revised manuscript, please respond to the comments made by the referee(s) and upload a file "Response to Referees" in "Section 6 - File Upload". You can use this to document any changes you make to the original manuscript. In order to expedite the processing of the revised manuscript, please be as specific as possible in your response to the referee(s).

Please see our detailed instructions for revision requirements
<https://royalsociety.org/journals/authors/author-guidelines/>

Sincerely,
The Open Biology Team
mailto: openbiology@royalsociety.org

Handling Editor's comments:

The overall assessment of current work is that the study is quite important and interesting, however several points still need to be addressed. These would involve further attempts to

demonstrate efficacy of new vaccine platform against OVA+ tumor cells, repeating experiments that demonstrate T cell activation against neo-antigen (Figure 5D), presenting biophysical characterization of DC15 and comment on its ability to cross-react with human MHC II (or reference if this has been already reported), addressing/commenting on whether DC15 also stimulate CD4 T cells via direct MHC class II loading, and clarifying/adding p values to panels in Figures 4 and 5.

Reviewer(s)' Comments to Author(s):

Referee: 1

Comments to the Author(s)

The authors link neoantigens to an anti-MHC class II nanobody, DC15, by sortagging, and test whether these novel DC15-neoantigens can boost CD8 T cell activities against pancreatic cancer and melanoma in mice. Although anti-tumor activities of these reagents are only moderately encouraging, the idea is still worth documenting. I support the publication of this paper, but have a few comments that may help the authors improve the data quality in the paper.

1. DC15-neoantigens are initially delivered to the class II MHC compartments and cross-presented to MHC class I molecules to activate CD8 T cells. This is supported by data shown in Figure 3B. I know $p=0.011$ is considered statistically significant; however, there are only three data points from MHCII KO mice immunized with DC15-SIIN, and one of the data point falls above the average of the five data points from similarly immunized C57BL/6 mice. The authors may consider to add a few more MHCII KO mice in this study. In fact, the authors should probably state how sample sizes in various mouse studies were determined.
2. The authors should provide statistical analyses and p values for all relevant panels in Figures 4 and 5. In Figure 5B, B16 melanoma cells grow faster in mice receiving DC15 M9 than in those receiving VHHcont M9. Is this statistically significant? If so, can the authors speculate what may have happened?
3. One single mouse was analyzed to obtain and present data shown in each flow panel in Figure 5D. The authors should repeat this experiment using appropriate mouse numbers and include statistical data to warrant data reproducibility.

Referee: 2

Comments to the Author(s)

Overall, the authors describe a tumor vaccination strategy that is innovative and novel. Their approach uses an MHC class II-targeting nanobody that is conjugated to a tumor peptide. Their system uses sortase to conjugate tumor peptides, making it a modular system that can be rapidly adapted for any tumor antigen or immunogenic peptide. They propose that their nanobody (DC15) binds to MHC class II on APCs and gets internalized, subsequently resulting in cross-presentation of the tumor antigen. The authors demonstrate the fundamental aspects of this strategy are valid in vitro. Unfortunately, the authors tested their vaccination strategy in two independent in vivo models and did not observe robust anti-tumor effects. They do, however, observe evidence of immune activation in vivo as a consequence of DC15 conjugated to a peptide. This is an important finding and I believe the manuscript can become outstanding with revision as described below.

Major

The manuscript would benefit from an experimental system showing definitive evidence of therapeutic efficacy in vivo. The authors show T cell stimulation in vivo using the OT-1 model. Therefore, one approach could be to evaluate anti-tumor responses with DC15-SIIN against an

OVA+ tumor. Alternatively, perhaps a more robust anti-tumor response could be observed when combining a DC15-peptide conjugate with an immune checkpoint inhibitor. Along these lines, Moynihan et al 2016 Nature Med (PMID 27775706) found that peptide vaccines were most effective when combined with IL-2 analogues and immune checkpoint blockade. At the very least, the experiment reported in Figure 5D should be repeated with a larger number of animals to draw statistical conclusions. If the authors approach is not effective in models of cancer, might it alternatively work as a strategy to immunize against intracellular pathogens?

Minor

1. Has the DC15 agent been described in the literature previously? If not, the authors should present its biophysical characterization with respect to target binding and affinity. Does the protein cross-react with human MHC class II?
2. Does DC15 also stimulate CD4 T cells via direct MHC class II loading?
3. The text could benefit from a diagram that demonstrates the proposed mechanisms of action of DC15-peptide conjugates.

Author's Response to Decision Letter for (RSOB-19-0235.R0)

See Appendix A.

RSOB-19-0235.R1 (Revision)

Review form: Reviewer 1

Recommendation

Accept as is

Do you have any ethical concerns with this paper?

No

Comments to the Author

The authors have addressed my concerns. I now support the publication of this paper in Open Biology.

Review form: Reviewer 2

Recommendation

Accept as is

Do you have any ethical concerns with this paper?

No

Comments to the Author

The authors have appropriately addressed the points raised by the reviewers. I recommend this manuscript be accepted without further revision.

Decision letter (RSOB-19-0235.R1)

06-Jan-2020

Dear Dr Dougan,

We are pleased to inform you that your manuscript entitled "Neoleukin-2 enhances anti-tumor immunity downstream of peptide vaccination targeted by an anti-MHC class II VHH" has been accepted by the Editor for publication in Open Biology.

If applicable, please find the referee comments below. No further changes are recommended.

As board member, all article processing charges are waived (applicable to one submission per year).

With best wishes for the new year,

The Open Biology Team
mailto:openbiology@royalsociety.org

Reviewer(s)' Comments to Author:

Referee: 2

Comments to the Author(s)

The authors have appropriately addressed the points raised by the reviewers. I recommend this manuscript be accepted without further revision.

Referee: 1

Comments to the Author(s)

The authors have addressed my concerns. I now support the publication of this paper in Open Biology.

Appendix A

Handling Editor's comments:

The overall assessment of current work is that the study is quite important and interesting, however several points still need to be addressed. These would involve further attempts to demonstrate efficacy of new vaccine platform against OVA+ tumor cells, repeating experiments that demonstrate T cell activation against neo-antigen (Figure 5D), presenting biophysical characterization of DC15 and comment on its ability to cross-react with human MHC II (or reference if this has been already reported), addressing/commenting on whether DC15 also stimulate CD4 T cells via direct MHC class II loading, and clarifying/adding p values to panels in Figures 4 and 5.

Thank you for this thoughtful summary. We have included new data showing that DC15 conjugated vaccines increase the frequency of tetramer+ CD8 T cells in mice post vaccination as Revised Figure 5D. We have also included new data showing MHC class II is required for the efficacy of DC15 based vaccination (Revised Figure 3B). The biophysical properties of DC15 have been previously reported, and we have revised the text accordingly. We have also added p values and made other minor changes as suggested by the reviewers below, including adding a summary diagram as Revised Figure 6.

Reviewer(s)' Comments to Author(s):

Referee: 1

Comments to the Author(s)

The authors link neoantigens to an anti-MHC class II nanobody, DC15, by sortagging, and test whether these novel DC15-neoantigens can boost CD8 T cell activities against pancreatic cancer and melanoma in mice. Although anti-tumor activities of these reagents are only moderately encouraging, the idea is still worth documenting. I support the publication of this paper, but have a few comments that may help the authors improve the data quality in the paper.

1. DC15-neoantigens are initially delivered to the class II MHC compartments and cross-presented to MHC class I molecules to activate CD8 T cells. This is supported by data shown in Figure 3B. I know $p=0.011$ is considered statistically significant; however, there are only three data points from MHCII KO mice immunized with DC15-SIIN, and one of the data point falls above the average of the five data points from similarly immunized C57BL/6 mice. The authors may consider to add a few more MHCII KO mice in this study. In fact, the authors should probably state how sample sizes in various mouse studies were determined.

We agree that showing a requirement for MHC class II expression is an important confirmation of the mechanism of action of the targeted DC15 vaccine. Our original figure showed mice vaccinated with 100 ng protein per mouse; however, at this very high dose we did occasionally see responses that were independent of MHC class II targeting, which we ascribed to protein conjugates being taken up by nonspecific endocytosis. We have therefore replaced Figure 3B with a different experimental replicate in which mice were vaccinated with 10 ng protein per mouse, a dose at which the requirement for MHC class II targeting is more obvious. We have also included 5 mice per group and report a p value of <0.001 .

2. The authors should provide statistical analyses and p values for all relevant panels in Figures 4 and 5. In Figure 5B, B16 melanoma cells grow faster in mice receiving DC15 M9 than in those

receiving VHHcont M9. Is this statistically significant? If so, can the authors speculate what may have happened?

We have added statistical comparisons to Figures 4 and 5. The DC15-M9 vaccinated mice in Figure 5B did have tumors that grew significantly faster than the VHHcont-M9 vaccinated mice. We speculate that this could be due to DC15 binding to B cells expressing MHC class II. However, in the therapeutic context the DC15-M9 vaccination performed better than VHHcont-M9. It could be that the inoculation of tumor itself acts as a complex mixture of vaccine antigens and that priming with a single peptide followed by boosting with a complex mixture is less effective than priming with a complex mixture and boosting with a single peptide. We have included additional text and an additional reference to the results section describing these findings.

3. One single mouse was analyzed to obtain and present data shown in each flow panel in Figure 5D. The authors should repeat this experiment using appropriate mouse numbers and include statistical data to warrant data reproducibility.

We agree that $n=1$ per group is not very many mice. Unfortunately, these were all of the mice that survived tumor-free at the end of both the prophylactic and therapeutic experiments. To get a better sense of the vaccine-elicited CD8 T cell response, we therefore decided to look at an earlier time point when all of the mice were still alive. We vaccinated mice according to the prophylactic schedule shown in Figure 5A and harvested draining lymph nodes on Day -5. At this time point, mice had received 2 footpad vaccinations and multiple doses of Neo2/15, but had not been inoculated with tumors. At this time point, we found that DC15-M9 elicited a modest, but statistically significant, increase in TRP1 tetramer+ CD8 T cells as compared to mice receiving VHHcont-M9 or DC15-SIIN. The magnitude of the response is somewhat weak, which is consistent with the tumor growth and survival curves that we observed. We have presented these data as **Revised Figure 5D** and included new text in the results and discussion section.

Referee: 2

Comments to the Author(s)

Overall, the authors describe a tumor vaccination strategy that is innovative and novel. Their approach uses an MHC class II-targeting nanobody that is conjugated to a tumor peptide. Their system uses sortase to conjugate tumor peptides, making it a modular system that can be rapidly adapted for any tumor antigen or immunogenic peptide. They propose that their nanobody (DC15) binds to MHC class II on APCs and gets internalized, subsequently resulting in cross-presentation of the tumor antigen. The authors demonstrate the fundamental aspects of this strategy are valid *in vitro*. Unfortunately, the authors tested their vaccination strategy in two independent *in vivo* models and did not observe robust anti-tumor effects. They do, however, observe evidence of immune activation *in vivo* as a consequence of DC15 conjugated to a peptide. This is an important finding and I believe the manuscript can become outstanding with revision as described below.

Major

The manuscript would benefit from an experimental system showing definitive evidence of therapeutic efficacy *in vivo*. The authors show T cell stimulation *in vivo* using the OT-1 model. Therefore, one approach could be to evaluate anti-tumor responses with DC15-SIIN against an OVA+ tumor. Alternatively, perhaps a more robust anti-tumor response could be observed when combining a DC15-peptide conjugate with an immune checkpoint inhibitor. Along these lines, Moynihan et al 2016 Nature Med (PMID 27775706) found that peptide vaccines were most effective when combined with IL-2 analogues and immune checkpoint blockade. At the very least,

the experiment reported in Figure 5D should be repeated with a larger number of animals to draw statistical conclusions. If the authors approach is not effective in models of cancer, might it alternatively work as a strategy to immunize against intracellular pathogens?

We fully agree with the reviewer that we would rather have had a more efficacious vaccine. We feel that in the interests of transparency we should show modest performance against actual tumor antigens rather than using the OVA model antigen. To better quantify the effect of our VHH-peptide vaccine, we analyzed TRP1 tetramer+ cells at an earlier time point in the vaccination response (9 days post first vaccination). At this point, we observed a statistically significant increase in antigen-specific CD8+ T cells from the DC15-M9 vaccine; however, the overall magnitude of the response was admittedly low. We now include these data as **Revised Figure 5D**. We also now cite Moynihan et al which is one of our favorites for its reliance on naturally occurring self-antigens as tumor rejection antigens.

MHC class II targeted VHHs have been used as vaccines for infectious disease in settings where CD4 T cell responses and antibody responses conferred protective immunity. They have thus far been less good at eliciting CD8 T cell responses, and we now include additional discussion on this topic.

Minor

1. Has the DC15 agent been described in the literature previously? If not, the authors should present its biophysical characterization with respect to target binding and affinity. Does the protein cross-react with human MHC class II?

Yes, DC15 was first reported in Rashidian et al. ACS 2015 (PMID: 26955657). In this report, the authors demonstrate specificity of the VHH by lack of staining on MHC class II-deficient B cells and co-staining of MHC class II-GFP B cells. DC15 was also shown to competitively inhibit binding of VHH7, a known anti-MHC class II VHH. The affinity of DC15 was calculated by titration of fluorescently labeled DC15 versus VHH7 in a flow cytometry-based assay. By this comparison, DC15 had a 5-fold higher affinity relative to VHH7. The affinity of VHH7 had been previously reported in PMID: 27821668 (K_D for I-A^b = 7.0 nM). Neither VHH7 nor DC15 cross-react with human MHC class II, although a pan-haplotype anti MHC class II does exist for humans (VHH4).

We have now included this information in the introduction and discussion sections of the revised manuscript.

2. Does DC15 also stimulate CD4 T cells via direct MHC class II loading?

VHH7 stimulates CD4 T cells via direct MHC class II loading as reported in Duarte et al. JI 2016 (PMID: 27821668). Given that the DC15 and VHH7 epitopes are overlapping, we predict that DC15 would similarly enhance antigen presentation on MHC class II, although we have not formally demonstrated this point in vivo.

3. The text could benefit from a diagram that demonstrates the proposed mechanisms of action of DC15-peptide conjugates.

We have now added a diagram as **Revised Figure 6**.